# Distinct Functional Alterations and Therapeutic Options of Two Pathological De Novo Variants of the T292 Residue of GABRA1 Identified in Children with Epileptic Encephalopathy and Neurodevelopmental Disorders

**DOI:** 10.3390/ijms23052723

**Published:** 2022-03-01

**Authors:** Wenlin Chen, Yang Ge, Jie Lu, Joshua Melo, Yee Wah So, Romi Juneja, Lidong Liu, Yu Tian Wang

**Affiliations:** 1Djavad Mowafaghian Centre for Brain Health and Department of Medicine, University of British Columbia, Vancouver, BC V6T 2B5, Canada; chenwenlin1991@gmail.com (W.C.); yang.ge@ubc.ca (Y.G.); jielu@mail.ubc.ca (J.L.); evaso@mail.ubc.ca (Y.W.S.); ytwang@brain.ubc.ca (Y.T.W.); 2Neurology Centre of Toronto, Toronto, ON M5N 1A8, Canada; joshua.melo97@gmail.com

**Keywords:** epileptic encephalopathy, neurodevelopmental disorders, de novo missense variants, GABRA1 T292 residue, channel gating properties, therapeutic options

## Abstract

Mutations of GABA_A_R have reportedly led to epileptic encephalopathy and neurodevelopmental disorders. We have identified a novel de novo T292S missense variant of GABRA1 from a pediatric patient with grievous global developmental delay but without obvious epileptic activity. This mutation coincidentally occurs at the same residue as that of a previously reported GABRA1 variant T292I identified from a pediatric patient with severe epilepsy. The distinct phenotypes of these two patients prompted us to compare the impacts of the two mutants on the receptor function and to search for suitable therapeutics. In this study, we used biochemical techniques and patch-clamp recordings in HEK293 cells overexpressing either wild-type or mutated rat recombinant GABA_A_Rs. We found that the α1T292S variant significantly increased GABA-evoked whole-cell currents, shifting the dose–response curve to the left without altering the maximal response. In contrast, the α1T292I variant significantly reduced GABA-evoked currents, shifting the dose–response curve to the right with a severely diminished maximum response. Single-channel recordings further revealed that the α1T292S variant increased, while the α1T292I variant decreased the GABA_A_R single-channel open time and open probability. Importantly, we found that the T292S mutation-induced increase in GABA_A_R function could be fully normalized by the negative GABA_A_R modulator thiocolchicoside, whereas the T292I mutation-induced impairment of GABA_A_R function was largely rescued with a combination of the GABA_A_R positive modulators diazepam and verapamil. Our study demonstrated that α1T292 is a critical residue for controlling GABA_A_R channel gating, and mutations at this residue may produce opposite impacts on the function of the receptors. Thus, the present work highlights the importance of functionally characterizing each individual GABA_A_R mutation for ensuring precision medicine.

## 1. Introduction

Epilepsy is a neurological disorder characterized by recurrent seizures that occur due to abnormally excessive electrical discharges of the cerebral neurons as a result of the disruption of the excitation–inhibition balance (E/I balance) in the brain [1,2,3]. About 50% of the epilepsies diagnosed worldwide are gene-related [4]. Recent studies have identified several hundred genes associated with epilepsy, among which genes encoding ion channels/receptors predominate [3,5]. As a Cl^−^ channel and the primary mediator of inhibitory synaptic neurotransmission in the central nervous system (CNS), the γ-aminobutyric acid (GABA) type A receptor (GABA_A_R) plays a critical role in maintaining the neuronal E/I balance in the CNS and is considered an important genetic risk factor for epilepsy. In addition, the dysfunction of the receptor underlies the pathogenesis of many neurological diseases. Accumulating evidence suggests that genetic variants of the GABA_A_R may cause conditions including epileptic encephalopathy (EE) [6,7,8,9], developmental delay [9,10], Fragile X, Rett Syndrome, and Dravet Syndrome [11,12,13].

Structurally, ionotropic GABA_A_Rs are heteropentamers of the α (α_1_–α_6_), β (β_1_–β_3_), γ (γ_1_–γ_3_), δ, ε, θ, π, or ρ subunits, with α and β being the obligatory subunits [9]. Most synaptic GABA_A_Rs are assembled from two α_1_ subunits, two β_2_ subunits, and one γ_2_ subunit [14]. Each subunit contains a large extracellular N-terminal domain, four transmembrane helices, two intracellular loops, and one extracellular C-terminal domain (Figure 1A) [15,16,17]. The receptor’s agonist, GABA, binds to the agonist-binding site interfacing the extracellular N-terminal domains of the α and β subunits (Figure 1B). The agonist binding triggers the conformational opening of the receptor channel and leads to chloride anion influx through the ion pore formed by the transmembrane segments of all five subunits [13,17,18]. Due to the pentameric composition of the GABA_A_R, mutations of the amino acid residues in many different GABA_A_R subunits have been reported to alter the function, expression level, and subcellular distributions of the receptor [19,20,21,22].

The α1 subunit of the GABA_A_R, which is encoded by the GABRA1 gene, is ubiquitously expressed in CNS neurons, suggesting its importance in maintaining the normal function of the vast majority of native GABA_A_Rs [23]. Indeed, increasing numbers of GABRA1 variants have been implicated in causing haploinsufficiency and loss of function of the GABA_A_Rs, thereby causally contributing to the pathogenesis of various forms of epilepsy [24,25,26,27,28], Dravet Syndrome, early-onset EEs, and developmental delay [13,29]. In nearly all patients with pathogenic GABRA1 variants, the clinical phenotypes include epilepsies within a spectrum of different severity, ranging from generalized epilepsies to severe epileptic encephalopathies [23].

We have recently identified a novel de novo T292S (*C875G*) missense variant of GABRA1 from a pediatric patient with developmental delay, but, to our surprise, without diagnosed seizure events. Most interestingly, this mutation occurs on the same residue as that of the previously reported variant T292I (*C875T*) identified from a pediatric patient with severe epilepsy [19]. The dramatic difference between the phenotypes of the two patients and the resistance to the standard care treatments of the patient harbouring the T292S mutation prompted us to characterize and compare the pathological impacts of the two mutations on the function of the GABA_A_R and potentially search for the most suitable and effective pharm-therapeutics. To accomplish our goals, we used surface biotinylation, Western blotting, and whole-cell and single-channel patch-clamp recordings to characterize the two variants in HEK293 cells overexpressing either the wild-type or mutant rat recombinant GABA_A_Rs containing the α1/β2/γ2 subunits. We found that the two mutations have drastically different impacts on the receptor function: The α1_T292S_ variant significantly increased, whereas the α1_T292I_ variant significantly reduced GABA_A_R function. The opposite functional impacts of these mutations suggest that we would need different therapeutic strategies for treating the patients carrying these mutations. Indeed, through screening several clinically approved drugs that reportedly act on GABA_A_Rs, we found that the GABA_A_R allosteric inhibitor thiocolchicoside (TCC) [30] could reduce the gain of function caused by the T292S mutation, thereby normalizing the receptor function. On the other hand, a combination of GABA_A_R positive modulators, diazepam and verapamil [18,20], largely rescued the loss of function caused by the T292I mutation. Our results, particularly the gain of function caused by α1_T292S_, are in great contrast to the loss of function of most, if not all, previously reported pathogenic GABRA1 mutations [19]. This highlights the importance of the functional characterization of and search for different therapeutic strategies for each individual mutation.

## 2. Results

### 2.1. Phenotypic Comparisons of the Two Patients Carrying the GABRA1 Mutants

Our patient was a 2-year-old boy with severe neurodevelopmental delays manifested as visual impairment, feeding difficulties, and significantly below-average body weight and organ mass. He had, however, no diagnosed overt seizure activity despite a few abnormal electroencephalography (EEG) discharges observed. In addition, there was no abnormality that could be found in MRI. No similar neurodevelopmental phenotype has been observed in his siblings or family members. Through whole-exome sequencing (WES), we identified a novel de novo T292S (*C875G*) missense variant of GABRA1. The clinical phenotypes, particularly the lack of any observable somatic seizure activity, are drastically different from that of a previously reported patient who carried a de novo variant at the same residue T292I (*C875T*) and had the main clinical features of Lennox–Gastaut syndrome with light-sensitive myoclonic epilepsy, generalized tonic–clonic seizures, and developmental delay [19,31]. These strikingly different clinical phenotypes may implicate distinct impacts on the receptor functions by these two mutations albeit occurring on the same residue.

### 2.2. Structural Analysis and the Location of T292 Residue in GABRA1

The α1. T292 residue is located in the second transmembrane domain (TM2) of the α_1_ subunit (Figure 1A) and lies within the inner channel pore (Figure 1B). The parents of both patients have normal alleles, but the patients have the heterozygous missense variant of the same site in GABRA1 (NM_000806.5: *C.875C>G* or *875C>T*, NP_000797.2: p.Thr292Ser or Thr292Ile; Figure 1C). Sequence alignment demonstrates that this residue is highly conserved among different species, including human, monkey, rat, mouse, bovine, chicken, xenopus, and zebrafish species, implying the importance of this residue (Figure 1D). This is in good agreement with previous studies reporting that modification or pathogenic variant of the channel-lining residues could lead to dramatic alterations of the receptor function, particularly channel-gating properties [32,33].

### 2.3. The T292S and T292I Variants Bidirectionally Affect GABA-Evoked Responses

The drastic differences in the clinical phenotypes of the two mutations may imply significantly different impacts on the GABA_A_R function. Since the functional impacts of neither variant have previously been fully characterized, we employed whole-cell patch-clamp recordings of GABA-evoked currents to test whether the variants would affect GABA_A_R functions. The evoked whole-cell currents were measured by fast perfusion (2 s) of GABA with concentrations ranging from 0.1 μM to 1 mM at a holding membrane potential of −60 mV in HEK293 cells transiently overexpressing wild-type α1/β2/γ2, α1_T292S_/β2/γ2, or α1_T292I_/β2/γ2.

For the wild-type GABA_A_Rs, GABA-evoked inward currents increased in a dose-dependent manner, in which the currents were observable at 1 μM and reached the maximum level at around 100 μM. In comparison with the wild-type receptor, the GABA sensitivity of the α1_T292S_ GABA_A_Rs was significantly increased (Figure 2). The currents gated by the mutated receptor were observable at the concentration of 0.1 μM of GABA and reached the maximal at around 10 μM of GABA (Figure 2A). When we normalized the currents to the cell’s own maximum response, we found that in comparison with the wild-type GABA_A_Rs, T292S GABA_A_Rs had shifted its dose–response curve to the left (Figure 2C) and lowered its EC50 by more than ten-fold (T292S: 0.2895 μM; WT: 3.658 μM) (Figure 2C,E). However, when we normalized the currents of the T292S GABA_A_Rs (3287 ± 422.5 pA) against the maximum response of WT (3075 ± 526.0 pA), we observed that the maximum response of the T292S GABA_A_Rs was not affected (Figure 2B,D). To mimic the patient’s heterozygous expression of the mutant subunit, we co-transfected both the wild-type α1_T292_ and the α1_T292S_ mutant in a 1:1 ratio along with the β2 and γ2 subunits in HEK293 cells. We found that this heterozygous expression caused a shift of the GABA-evoked dose–response curve toward the midpoint between the wild-type and the homozygous curves (Appendix A).

In stark contrast, the expression of the T292I mutant resulted in significant impairment of GABA_A_R function. It severely reduced the peak current amplitude at each concentration tested (Figure 2A) with a drastically decreased maximum response (1033 ± 286.5 pA, which was only about one-third of the response of the wild-type or T292S GABA_A_Rs) (Figure 2B,D). The reduced maximum response was also associated with a rightward-shifted dose–response curve and a more than 18-fold increased EC50 over the wild-type GABA_A_R (T292I: 69.28 μM; WT: 3.658 μM) (Figure 2C–E).

To check whether either of the two variants could affect the ion selectivity of GABA_A_R, chloride ion channel amplitudes of the GABA-evoked peak currents were measured at stepwise membrane potentials ranging from −80 to +80 mV with 20 mV intervals. As shown in Figure 2F,G, we found that neither variant compromised the ion selectivity since the reversal potentials of both variant receptors were similar to that of wild-type receptors, in which all were at approximately 0 mV.

### 2.4. Neither T292S nor T292I Variants Affected GABA_A_R Total/Surface Expressions

The marked functional alterations of the two variants in the GABA-evoked currents were quite intriguing and prompted us to probe the mechanisms underlying the bidirectional changes. Functional changes could be due to changes in receptor expression or channel-gating properties. Since alteration in the level of GABA_A_Rs expression in the cell and/or on the cell membrane, including other previously identified GABA_A_R variants [23,34,35], have previously been reported to be important methods of regulating the functions of the receptors with physiological and pathogenic consequences, we first examined the possibility that these two variants would exert their impacts on the receptor by affecting the receptor total and/or surface expression levels. We quantified the total receptors expressed in the cell by immunoblotting total cell lysates and quantified the surface receptors expressed on the plasma membrane by specifically immunoblotting biotinylated surface receptors (Figure 3) in HEK cells transiently expressing wild-type α1/β2/γ2, α1_T292S_/β2/γ2, or α1_T292I_/β2/γ2. To our surprise, unlike other previously identified GABA_A_R variants [19,20,22], we found that in comparison with the WT receptors, neither variant produced any significant alteration in either the total or the surface expression levels of the receptor (Figure 3A–D).

### 2.5. T292S and T292I Variants Bidirectionally Altered GABA_A_R Single Channel Properties

Since neither T292S nor T292I mutation affected GABA_A_R expression in the cell or on the plasma membrane as shown above, we next examined the possibility that the functional alterations resulted from the changes in channel-gating properties. We employed cell-attached, single-channel recordings of the WT and mutant GABA_A_Rs transiently expressed in HEK cells with subsaturating (3 μM) or saturating (1 mM) concentrations of GABA contained in the recording pipette. As shown in Figure 4A–E, we found that the T292S mutant significantly increased GABA-evoked single-channel activity at subsaturating, but not at saturating, GABA concentrations. When induced by 3 μM of GABA, the T292S variant significantly increased the mean open time (T292S: 31.22 ± 3.532 ms vs. WT: 16.58 ± 0.4146 ms), and hence the open probability (T292S: 23.58 ± 4.279% vs. WT: 8.528 ± 1.34%) of GABA_A_R without changing the channel conductance (T292S: 29.16 ± 1.492 pS, WT: 29.51 ± 1.806 pS) and channel open frequency (T292S: 7.982 ± 1.719 s^−1^, WT: 5.172 ± 0.8122 s^−1^) in comparison with the WT GABA_A_R (Figure 4A–E), which was in good agreement with the increased sensitivity to GABA. In contrast, the T292I mutant significantly reduced sensitivity to GABA in comparison with the WT receptors as no detectable single-channel event under subsaturating concentrations of GABA (Figure 4A–E) was observed. When the currents were evoked with GABA at a concentration of 1 mM, the T292S GABA_A_Rs showed no significant alterations of conductance (T292S: 28.27 ± 2.199 pS vs. WT: 24.76 ± 1.095 pS), mean open time (T292S: 33.77 ± 5.447 ms vs. WT: 30.36 ± 4.247 ms), open frequency (T292S: 10.7 ± 2.358 s^−1^ vs. WT: 8.403 ± 2.057 s^−1^), or open probability (T292S: 30.19 ± 4.249% vs. WT: 22.93 ± 5.094%) (Figure 4F–J), which was in good agreement with the no observed change in the maximum response in the whole-cell current recordings mentioned above. (Figure 2). As expected, the T292I variant significantly reduced the single-channel activity evoked by the high concentration of GABA (1 mM). In particular, while no change in the conductance (T292I: 24.71 ± 1.396 pS, WT: 24.76 ± 1.095 pS) could be detected, both the channel-opening frequency (T292I: 4.806 ± 1.234 s^−1^, WT: 8.403 ± 2.057 s^−1^) and open probability (T292S: 7.502 ± 2.224%, WT: 22.93 ± 5.094%) were significantly reduced, leading to a reduced mean open time (T292I: 15.41 ± 2.251 ms, WT: 30.36 ± 4.247 ms) when compared with the recordings from the WT counterparts (Figure 4F–J). The results imply that the impaired function of the GABA_A_Rs by the T292I mutation can be attributed to the decreased open probability and opening frequency of the channel.

### 2.6. T292S and T292I Variants Resulted in Different Changes of GABA_A_R-Gated Tonic and Leak Currents

We noticed in a recent study by Butler, K.M., et al. that a T292K variant of the α_2_ subunit caused the tonic opening of GABA_A_Rs [22]. Since the α_1_ and α_2_ subunits are highly homologous, we wondered whether the T292S and T292I variants in the α1 subunit would cause any changes to the tonic/leak currents of GABA_A_Rs. The GABA tonic current is the current gated through the GABA_A_R that is activated by the ambient GABA in the extracellular solution, and it can be revealed by blocking the GABA_A_R with the competitive antagonist bicuculline [36]. The leak current is the current gated through the channel pore of the non-activated GABA_A_R and is usually revealed with a receptor channel pore blocker, such as picrotoxin [37]. We found that 20 μM of bicuculline was already enough to block all tonic current because the application of a higher concentration (50 μM) did not induce a further blockade (Appendix A). Bath application of the saturated bicuculline (20 μM) produced pronounced tonic currents as evident by the upward baseline current drifting in cells expressing T292S GABA_A_Rs (249 ± 52.6 pA). In great contrast, bicuculline application resulted in only minimal or no tonic currents in cells expressing either the WT (13.63 ± 5.641 pA) or T292I (0.3709 ± 0.8945 pA) receptors (Figure 5A,B). Similarly, bath application of picrotoxin (100 μM) revealed a notable leak current as evident with a small additional increase upward current on top of the bicuculline-induced tonic currents only in cells expressing the T292S GABA_A_Rs (72.22 ± 16.44 pA), but not in cells expressing the WT (1.586 ± 2.496 pA) or T292I (1.913 ± 2.304 pA) GABA_A_Rs (Figure 5A,C). These results demonstrated that the T292S variant could increase both tonic and leak currents of GABA_A_Rs while the T292I variant could not affect either current.

### 2.7. Thiocolchicoside Restores the Function of T292S GABA_A_R

The increased GABA sensitivity of T292S variant is a surprising result since it suggests that a gain-of-function mutant of the GABA_A_R is also pathogenic. We reasoned that negative GABA_A_R modulators may be able to normalize the function of the mutant receptor back to the WT receptor level, thereby exhibiting therapeutic potential to treat patients carrying this gain-of-function GABA_A_R variant. We examined the effect of three clinically approved negative allosteric GABA_A_R modulators, bemegride [38], flumazenil [39], and thiocochicoside (TCC) [30], on the whole-cell currents evoked with perfusion of a range of GABA from 0.1 μM to 1 mM. We found that among the three drugs tested at a concentration of 200 μM, only TCC was able to significantly reduce the increased GABA currents gated through the T292S variant (Figure 6A–D). We then tested the dose–response curve of TCC in decreasing T292S GABA_A_R gated currents evoked by 1 μM GABA in an effort to find an optimal concentration of TCC that was capable of fully restoring the GABA-sensitivity of T292S GABA_A_R (Figure 6A,E and Appendix A). As shown in Figure 6A,E, we found that 1 μM TCC was able to fully shift the dose–response curve leftward to the level of the WT GABA_A_R. 

### 2.8. Combination of Verapamil and Diazepam Partially Rescues the Function of T292I GABA_A_R

Since the T292I variant’s loss-of-function is primarily a result of reduced channel-opening probability, we reasoned that its channel abnormality may be restored by some positive GABA_A_R allosteric modulators, such as diazepam and verapamil, that have previously been shown to restore the function of some pathogenic loss-of-function GABA_A_R variants via improving channel-gating properties [20]. We first pretreated HEK293 cells transiently expressing the α1_T292I_/β2/γ2 receptors with verapamil (4 μM) for 24 h, as well as acutely applied verapamil during the recordings of the full dose response of the GABA-evoked currents. As shown in Figure 7A,C, verapamil showed slight rescuing effects of the T292I GABA_A_R. We repeated the experiment using diazepam [40] and found that it also only showed partial rescuing effects (Figure 7A,B). Then, we considered the possibility that a combination of verapamil and diazepam would result in a synergistic rescuing effect. Indeed, acute application of diazepam and verapamil together after chronic treatment of verapamil showed the best effect in shifting the GABA-evoked dose–response curve toward that of the WT counterpart (Figure 7A,D), although the shift remained incomplete.

## 3. Discussion

In this study, we reported a novel de novo missense variant T292S of the T292 residue of GABRA1 and compared it with a previously found de novo missense variant T292I of the same residue. As summarized in Table 1, clinical observations and laboratory investigations revealed drastically distinct clinical phenotypes and functional alterations in these GABA_A_Rs mutants. The patient carrying the T292S variant is featured with developmental delay without observable somatic seizure activity until deceased (while the possibility that epileptic activity could develop at a later developmental stage cannot be entirely ruled out); while the patient with the T292I variant primarily manifested with severe epilepsy [19,31]. Considering the fact the T292 residue is highly conserved among species and is one of the residues lining the channel pore, functional characterization of these variants identified from patients may improve our understanding of the underlying pathophysiology. Our functional analysis performed in HEK293 cells showed that the T292S and T292I variants of the GABA_A_R α1 subunit conferred opposing changes in GABA agonist sensitivity and potency. The T292S variant induced a leftward shift of the GABA dose–response curve and lowered the EC_50_ of GABA without altering the maximum response. On the other hand, the T292I variant caused a rightward shift of the GABA dose–response curve, increased the EC_50_ of GABA, and reduced the maximum response. To our knowledge, this is the first study that performs detailed functional characterizations of the T292S and T292I variants, and importantly, the first to report that the same subunit residue of GABA_A_R replaced by different amino acids would result in opposite impacts on the receptor functions.

There are several potential reasons that can explain these functional changes, which were examined in detail in our experiments [22]. First, the genetic mutations observed in the GABARA1 variants may theoretically alter the receptor expression, assembly, and trafficking that lead to the change of the surface/total receptor numbers [20]. However, we found that neither T292S nor T292I variants caused changes in the receptor numbers as evidenced by the results of immunoblotting and surface biotinylation assays.

Second, single amino acid mutations may induce structural changes in the GABA agonist-binding pocket and result in increased or decreased GABA binding affinity [22,24,41]. However, as the T292 residue is located at the inner channel pore, which is located far from the GABA binding region, the chance that the mutation causes an allosteric change in the structure of the GABA agonist-binding pocket is low.

Third, since this residue lies in the middle of the channel pore, its mutations may produce conformational changes that lead to altered GABA sensitivity in inducing channel gating [24,28]. Supporting this conjecture, our single-channel recording data showed significant changes in channel-gating properties in both variants. The T292S variant increased the single-channel open time and open probability under subsaturating (3 μM) and saturating (1 mM) GABA concentrations, suggesting the alteration of GABA_A_R function by increasing the sensitivity of GABA to induce channel opening. In contrast, the T292I variant showed a significant decrease in GABA’s ability to keep the channel opened as evident by the lack of opening activity at the subsaturating GABA level, and significantly reduced channel open time and open probability at the saturating GABA stimulation (1mM). Together, these results strongly suggest that residue T292 in the α1 subunit has a critical role in determining the channel open threshold by its agonists. The idea that the opposing functional alterations of these two mutations primarily resulted in changing GABA’s sensitivity in gating the receptor channel is further supported by the results of our tonic and leak currents assay. We found that the T292S (but not T92I) GABA_A_R showed increased tonic currents and leak currents. We postulated that the T292S variant not only decreases the threshold of GABA-induced channel opening, but also keeps the channel at a partially opened status even in the absence of agonist binding, whereas the T292I variant turns the channel into a more closed status by decreasing the channel opening ability. Given that the T292 residue is located at the TM2 segment and forms part of the channel pore, it should not be surprising to see such changes in the channel-gating properties. In particular, the T292I variant involves a threonine to isoleucine mutation, which may result in a significant change in the side chain’s length and polarity that are not in favour of channel opening. Similarly, as the T292S variant involves a more subtle threonine to serine mutation, it results in less alteration in the side chain’s length and polarity, and such relative subtle changes may, in turn, favour channel opening. Together, the ability of both T292S and T292I to dramatically and oppositely change the channel open probability demonstrates that α1T292 is a critical residue for controlling GABA_A_R channel gating.

Since we identified the first GABRA1 mutation, A322D, in an epilepsy patient two decades ago [42], there have been more than 25 GABRA1 mutations that are related to pediatric encephalopathy reported, such as T20I, V74I, S76R, F104C, R112Q, N115D, L146M, T156C, P181S, D219N, P260L, S270H, V287L, T292I, K306T A322D, A332V, as well as R214H/C, G251S/D, M263T/I, and T289P/A, in which the same residue is replaced by different amino acids [8,13,21,25,35,43]. Most of these characterized mutations cause a loss of function of the GABA_A_R due to reduced surface/total expression, decreased agonist sensitivity, or compromised channel gating [20,42,44,45]. In addition to our T292S variant reported here, only one of these GABRA1 variants, A332V, which is located in the channel pore-forming TM3, is recently reported to enhance the receptor function [41]. In the present work, we report for the first time that the T292S variant, unlike most of the previously reported pathogenic GABA_A_R variants, is clearly a gain-of-function variant; while the T292I variant, which occurs at the same residue with a different amino acid substitution as T292S, is a loss-of-function variant.

Following functional profiling of the pathological mechanisms of the T292S and T292I mutations, we also attempted to identify potential therapeutic options for more effective and personalized treatments. Guided by the detailed functional phenotypes of the two variants, we performed a quick screening of some clinically approved drugs that directly or indirectly act on GABA_A_Rs that may potentially reduce the functional abnormalities of certain mutants of GABA_A_Rs. Thiocochicoside (TCC) is used in the clinic as a muscle relaxant, but it is also an allosteric GABA_A_R inhibitor that shows potent antagonistic effects against GABA_A_R [30]. At a 1 μM concentration, TCC could fully restore the increased GABA-evoked dose response of the T292S variant to the WT level, with superior effects over other GABA_A_R negative allosteric modulators, bemegride and flumazenil. Our study, along with other previous studies, suggest that TCC can be a potential therapeutic option for gain-of-function GABA_A_R variants [22,41].

In contrast, we found that the loss-of-function variant T292I GABA_A_R-gated currents could be partially restored by a combination of two positive GABA_A_R modulating drugs, diazepam and verapamil. Chronic treatment of verapamil was shown to enhance channel gating with an elongated open time and increased open probability of R214C GABA_A_R, suggesting diazepam and verapamil may work synergistically to ultimately improve the function of the T292I variant GABA_A_Rs. Our study indicates that these mutations have opposite impacts on the function of GABA_A_Rs, and although occurring at the same residue, thereby require different functional and pharmacological strategies to restore the function of the receptor to the level of their WT counterpart. Thus, our findings suggest that it is of paramount importance to perform functional and pharmacological analysis after exome sequencing to determine the pathological mechanisms. This would aid in searching for the appropriate therapeutic options for patients carrying a de novo mutation of principle neurotransmitter receptors, such as the GABA_A_Rs.

In conclusion, our functional analyses of the two de novo variants, T292S and T292I, of the same GABRA1 T292 residue from two patients with distinct clinical phenotypes have revealed their gain-of-function and loss-of-function impacts on the GABA_A_R, respectively. The study not only provides evidence for the pathogenic contributions of the two variants to the patients’ pathology, but also indicates the crucial role of the T292 residue in controlling channel gating. There are around 30% of children with refractory epilepsy that do not respond to conventional drug treatments due to unknown functional alterations or unidentified causes [46,47]. Moreover, traditional anti-seizure drugs show little to no effect toward rescuing deficits caused by specific mutations in the GABA_A_R subunits [48,49], and in certain cases, can even exacerbate the symptoms [50,51]. The pharmacological characterizations in our study provide differential therapeutic suggestions for managing these two patients and endorse the importance of precision medicine for pediatric channelopathies.

## 4. Materials and Methods

### 4.1. Structural Modeling of GABA_A_ Receptor

The crystal structure of the GABA_A_R was obtained from the Protein Data Bank (PDB) (ID: 6X3U) [52]. Molecular graphics and analyses were performed with the PyMOL Academic-Professional version (Schrödinger, Inc., New York, NY, USA).

### 4.2. Plasmid Construction

The cDNAs encoding rat GABA_A_R α1, GABA_A_R β2, GABA_A_R γ2, and EGFP were separately cloned into the pcDNA3.0 vector (Invitrogen, Waltham, MA, USA). Using the plasmid with the WT rat GABA_A_R α1 cDNA as the template, PCR mediated QuickChange site-directed mutagenesis was performed with a high-fidelity Hot Start DNA polymerase (KAPA Biosystmems, Wilmington, MA, USA Cat: KM2605) to construct the mutated variants of the GABA_A_R α1 (c.875 C>G and c.875 C>T) subunit (point mutation primers: Sense 5′-cgaccgttctgagcatgacaacctt-3′, reverse 5′-aaggttgtcatgctcagaacggtcg-3′ and sense 5′-cgaccgttctgatcatgacaacctt-3′, reverse 5′-aaggttgtcatgatcagaacggtcg-3′ for T292S and T292I mutations, respectively). The mutations were confirmed by sequencing, and the resulting plasmids encoding the mutant GABA_A_R α1 were referred to as α1_T292S_ (T292S) and α1_T292I_ (T292I).

### 4.3. Cell Culture and Transfection

HEK293 cells were cultured in Dulbecco’s Modified Eagle Medium (DMEM; MilliporeSigma, St. Louis, MO, USA) supplemented with 10% fetal bovine serum (FBS; Invitrogen, Waltham, MA, USA). The cells were maintained at 37 °C with 5% CO_2_. For Western blot experiments investigating the expression of the GABA_A_R α_1_ subunits, cells were grown to 70% confluency in a poly-L-lysine-coated 6-well plate and co-transfected with plasmids encoding α1:β2:γ2 (1 μg:1 μg:0.5 μg), α1(T292S):β2:γ2 (1 μg:1 μg:0.5 μg), or α1(T292I):β2:γ2 (1 μg:1 μg:0.5 μg). The transfection was performed using the lipofectamine 2000 reagent (Invitrogen, Waltham, MA, USA) according to the manufacturer’s instructions. Transfected HEK293 cells were cultured for 40 h and then used for downstream Western blotting analysis and surface biotinylation assays. For electrophysiology experiments, cells were grown to 70–90% confluency in 6-well plates. EGFP (0.2 μg) plasmids were co-transfected with the GABA_A_R subunits to help visualize the successfully transfected cells during electrophysiology recording. Cells were maintained in 6-well plates for 5 h before being re-plated onto 12 mm glass coverslips coated with poly-L-lysine and were cultured for an additional 18–30 h before recording.

### 4.4. Western Blot and Surface Biotinylation

Transfected HEK293 cells in a 6-well cell culture plate were washed with ice-cold PBS three times and lysed with 0.5 mL of 1% SDS TBS buffer containing a protease inhibitor cocktail (Bimake, Houston, TX, USA) at 4 °C for 30 min. Cells were then harvested and homogenized using needles with gauge sizes from 18 G to 23 G to 26 G, progressively. The supernatant was collected after centrifugation (13,000 rpm, 4 °C, 10 min) and the protein concentration in each sample was measured using the Pierce BCA protein assay (Thermo Scientific, Waltham, MA, USA, REP233228). Samples containing equal amounts of the total protein were treated with the 6X Laemmli sample buffer containing 9% beta-mercaptoethanol (MilliporeSigma, St. Louis, MO, USA) and heated at 55 °C for 5 min before loading onto 10% SDS-PAGE gels. The proteins in the gels were transferred onto a PVDF membrane (MilliporeSigma, St. Louis, MO, USA). Rabbit anti-GABA_A_Rα1 polyclonal antibody (1:1000, MilliporeSigma, St. Louis, MO, USA, Cat. #06-868) was used to detect the WT and variant GABA_A_R α1 subunits. HSP90 (mouse monoclonal antibody 1:4000, BD, Biosciences, Franklin Lakes, NJ, USA, Cat. #610418) served as the loading control. The HRP-conjugated anti-mouse or anti-rabbit secondary antibodies (Thermo-Fisher Scientific, Waltham, MA, USA, Cat. # 31430 and 31460) were used in a ratio of 1:5000. The signal was detected using an ECL detection system (MilliporeSigma, St. Louis, MO, USA, Immobilon Crescendo Western HPP Substrate, Cat#WBLUR0500) via Bio-Rad ChemiDoc MP imaging system.

In the surface biotinylation assay, cells in a 6-well cell culture plate were washed 3 times on ice with ice-cold PBS 40 h post-transfection, then 0.5 mL of PBS with the membrane-impermeable reagent EZ-Link Sulfo-HNS-LC-Biotin (1 mg/mL, Thermo Scientific, Waltham, MA, USA) was added into each well and kept at 4 °C for 30 min to label surface membrane proteins. To quench the biotin reaction, cells were washed on ice with 1 mL of 100 mM glycine dissolved in ice-cold PBS three times (10 min each time) at 4 °C. Then the biotinylated cells in each well were harvested using 0.5 mL of 1% SDS lysate buffer (1% SDS in 1XTris Buffered saline, pH 7.6)) at 4 °C for 30 min and homogenized using needles with a gauge size from G18 to G23 to G26, progressively. Supernatants were collected after centrifugation (13,000 rpm, 4 °C, 10 min). The protein concentration in each sample was measured using the Pierce BCA protein assay (Thermo scientific, Waltham, MA, USA REP233228). Each sample with an equal amount of biotin-labeled membrane proteins was rotationally incubated with 30 uL of High-Capacity Streptavidin Agarose Resin (beads) (Thermo Scientific, Waltham, MA, USA REF20359) at 4 °C overnight, after which the beads were pulled down and washed with PBS for 3–4 times via centrifugation (500 g, 4 °C, 2 min). The beads were suspended with 45 uL of 2X Laemmli sample buffer and heated at 55 °C for 5 min. Heated samples were then centrifuged at 1000 rpm for 2 min. Finally, the supernatants were loaded onto 10% SDS-PAGE gels. Na^+^/K^+^ ATPase (abcam: Mouse monoclonal antibody alpha1 sodium potassium ATPase ([464.6] 1:1000) served as a loading control for the biotinylated membrane proteins and β-actin (MilliporeSigma, St. Louis, MO, USA: Mouse monoclonal antibody 1:3000, Cat, #A2228) served as a labeling control for the cytoplasmic proteins.

### 4.5. Electrophysiology

Electrophysiology experiments, including whole-cell voltage-clamp recordings and cell-attached single-channel recordings, were performed at room temperature on HEK293 cells transfected with the WT, T292S, or T292I GABA_A_R subunits. The recordings were conducted and low-pass-filtered at 2 kHz using a MultiClamp 700A amplifier (Molecular Devices, San Jose, CA, USA) and digitized at 20 kHz using Digidata 1440A. The recordings were performed using Clampex 10.7 software (Molecular Devices, San Jose, CA, USA) and the data were analyzed offline using Clampfit 10.7 (Molecular Devices, San Jose, CA, USA).

Whole-cell voltage clamp recordings were performed using an extracellular recording solution containing (in mM) 140 NaCl, 5.4 KCl, 2 MgCl_2_, 1.3 CaCl_2_, 10 HEPES, and 33 glucose (pH = 7.4, 310–320 mOsm). The thin-walled borosilicate glass patch pipettes (World Precision Instruments, Sarasota, FL, USA) were pulled to 3–5 MΩ resistance using a model P-97 micropipette puller (Sutter Instruments, Novato, CA, USA). During the recording, the glass patch pipettes were filled with an internal solution containing (in mM) 140 CsCl, 0.1 CaCl_2_, 2 MgCl_2_, 10 HEPES, 0.5 EGTA, and 4 ATP (K) (pH  =  7.2, 290–300 mOsm). For the GABA dose–response curve, the holding voltage was set at −60 mV, and GABA_a_R-gated currents were induced by applying various concentrations of GABA (0.1 μM–1000 μM, 2 s) using a two-barrel fast step perfusion system (Warner Instruments, Hamden, CT, USA). For the GABA_A_R current–voltage (I/V) relationship experiment, the holding voltages were set at (in mV) −80, −60, −40, −20, 0, +20, +40, +60, and +80 while applying a high concentration of GABA (1 mM, 1 s).

Tonic and leak currents assay was performed using a bath solution exchange protocol. Specifically, cells were patched and then recorded using a gap-free program, where 10 μM GABA, GABA-free ECS, 20 μM bicuculline, and 20 μM bicuculline plus 100 μM picrotoxin were bath applied sequentially and the application of each solution lasted at least 60 s to allow enough time for solution exchanges in the chamber. The initial application of GABA was meant to confirm the success of transfection of the recombinant GABA_A_Rs.

Cell-attached single-channel recordings were conducted using the same protocol as before [15]. The patch electrodes were fire-polished to a resistance of 10–20 MΩ and filled with the extracellular recording solution with 3 μM or 1 mM GABA (pH = 7.4, 310–320 mOsm), and the holding potential was +100 mV. Single-channel events were detected using the 50% amplitude threshold detection method and were visually inspected before being accepted. Single-channel open probability was determined by the fraction of open time over the total amount of analyzed time (120 s for each recording), and the mean channel open time was determined by the total amount of open time divided by the number of channel-opening events.

### 4.6. Chemicals

Picrotoxin (Abcam, Cambridge, UK), bemegride (Cedarlane, Burlington, ON, Canada), diazapem (Tocris Bioscience, Bristol, UK), and thiocolchicoside (Cedarlane, Burlington, ON, Canada) were first dissolved in DMSO for the stock solutions (100 mM) and further diluted in ECS to the desired working concentration during the experiment. Bicuculline (Hellobio Inc, Princeton, NJ, USA) and flumazenil (Tocris Bioscience, Bristol, UK) were first dissolved in DMSO for the stock solutions (20 mM) and further diluted in ECS during the experiment. Verapamil hydrochloride (Tocris Bioscience, Bristol, UK) was diluted in water to make a 4 mM stock solution.

### 4.7. Data Analysis

Data were presented as mean  ±  SEM (*n*  =  number of cells). The unpaired *t*-test, one-way ANOVA, and two-way ANOVA (both ANOVAs were followed by Bonferroni post hoc tests) were used for statistical analysis, and * *p*  <  0.05, ** *p*  <  0.01, and *** *p*  <  0.001 were considered statistically significant. Dose–response curves were fitted by the Hill equation, and EC_50_ was calculated by GraphPad prism 6 (GraphPad Software, San Diego, CA, USA). All data points were normalized to the maximum response of GABA-evoked current for each cell unless specified in the context. Whole-cell peak currents, the area under the curve, channel kinetics, and single-channel currents were analyzed by Clampfit 10.7(Molecular Devices, San Jose, CA, USA).

## Figures and Tables

**Figure 1 ijms-23-02723-f001:**
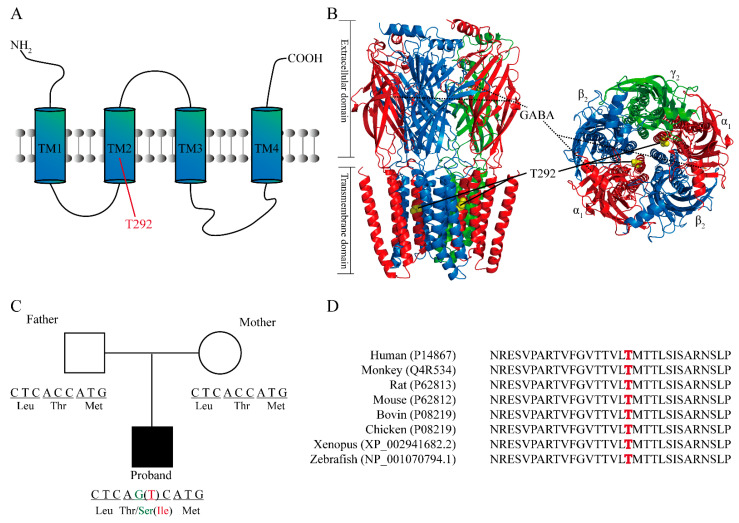
Structural analysis and the location of the T292 residue in GABRA1 and depiction of the de novo missense variants *c.875C>G* (p.Thr292Ser) (green) and *c.875C>T* (p.Thr292Ile) (red). (**A**) Diagram represents the T292 residue located at GABRA1 TM2. (**B**) Cryo-EM structure of the GABA_A_ receptor (PDB ID 6X3U) with highlighted T292 residue (yellow) and the bond agonist GABA (orange). As indicated, the T292 residue forms part of the inner channel pore. (**C**) Depiction of the two de novo missense variants in the patients and unaffected parents. (**D**) Comparison of the GABRA1 protein from several species indicates that Thr292 (in bold red) and nearby amino acids are evolutionarily conserved. Protein sequences were acquired from Uniprot and Ensembl.

**Figure 2 ijms-23-02723-f002:**
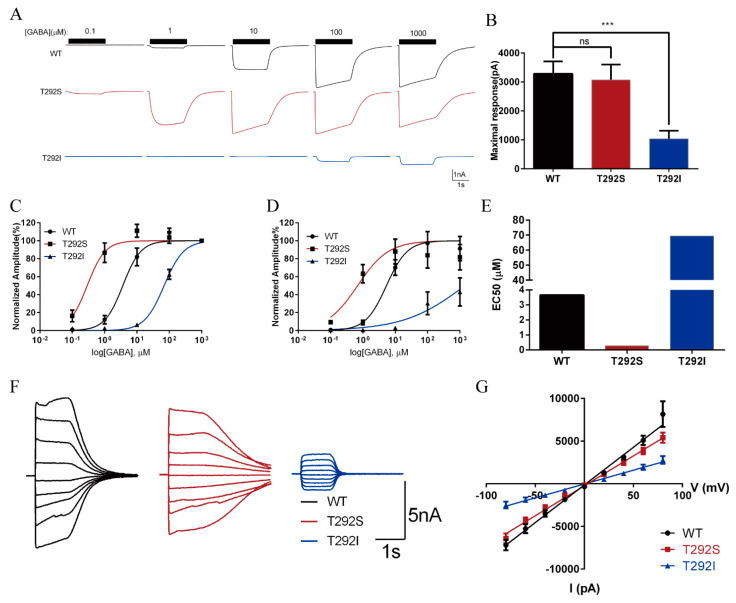
GABA-evoked responses of WT (black), α1_T292S_ (red), and α1_T292I_ (blue) of GABA_A_R show bidirectional changes in two de novo mutations at the same T292 residue of GABRA1. (**A**) Representative traces of GABA-evoked currents of WT, α1_T292S_, and α1_T292I_ of GABA_A_R. GABA application (2 s) is indicated as a bold black line above the traces. (**B**) Maximum response of α1_T292S_ (*n* = 10) and α1_T292I_ (*n* = 12) of GABA_A_R in comparison with WT (*n* = 8). (**C**) Dose-response curves of GABA-evoked responses of WT (*n* = 8), α1_T292S_ (*n* = 10), and α1_T292I_ (*n* = 12) of GABA_A_R. The peak current amplitude at each GABA concentration was normalized to the maximum response (1 mM) of each receptor, respectively. (**D**) Dose-response curves of GABA-evoked responses of WT (*n* = 8), α1_T292S_ (*n* = 10), and α1_T292I_ (*n* = 12) of GABA_A_R. The peak current amplitude at each GABA concentration was normalized to the maximum response (1 mM) of WT. (**E**) EC50 of GABA-evoked response of WT, α1_T292S_, and α1_T292I_ GABA_A_R. (**F**) Representative trace of I-V curve. (**G**) I-V curves for WT (*n* = 5) and mutant (*n* = 5) GABA_A_R-gated currents evoked by 1mM GABA. Statistical differences were determined using Student’s *t*-test (*** *p* < 0.001, ns = not significant).

**Figure 3 ijms-23-02723-f003:**
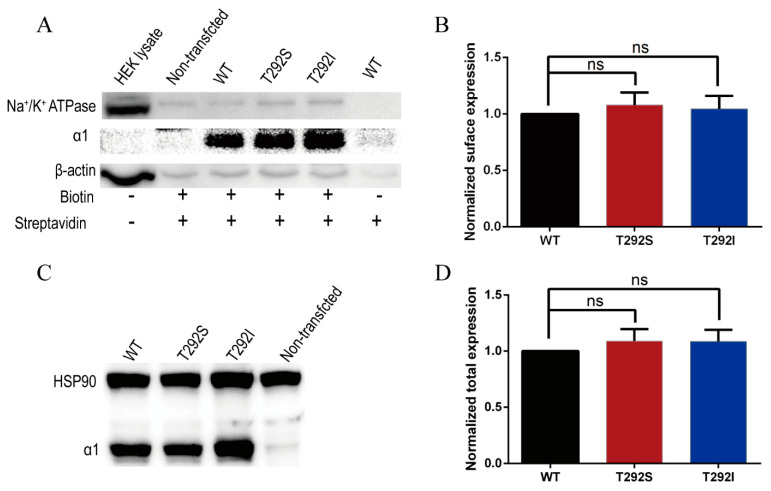
T292S and T292I variants did not affect GABA_A_R total/surface expressions. (**A**) Representative blots of surface biotinylation of WT, α1_T292S_, and α1_T292I_ of GABA_A_R and non-transfected HEK293. (**B**) Quantification of the surface α1 subunit expression normalized to Na^+^/K^+^ ATPase (*n* = 9). (**C**) Representative blots of the total expression levels from WT, α1_T292S_, and α1_T292I_ of GABA_A_R and non-transfected HEK293 samples. (**D**) Quantification of the total α1 subunit expression normalized to heat shock protein 90 (HSP90) (*n* = 7). Statistical differences were determined using one-way ANOVA (ns = not significant).

**Figure 4 ijms-23-02723-f004:**
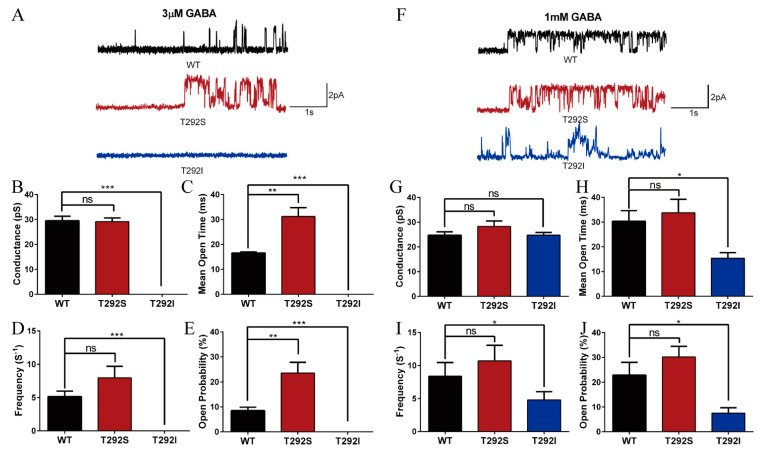
T292S and T292I variants bidirectionally altered GABA_A_R single-channel properties. Cell-attached single-channel currents were recorded for WT (black), α1_T292S_ (red), and α1_T292I_ (blue) of GABA_A_R under subsaturating and saturating concentrations of GABA. (**A**) Representative trace of α1_T292S_ (*n* = 6), α1_T292I_ (*n* = 6), and WT (*n* = 6) GABA_A_R under 3 μM GABA. (**B**) Conductance under 3 μM GABA condition. (**C**) Mean open time under 3 μM GABA condition. (**D**) Channel-opening frequency under 3 μM GABA condition. (**E**) Open probability under 3 μM GABA condition. (**F**) Representative trace of α1_T292S_ (*n* = 6), α1_T292I_ (*n* = 6), and WT (*n* = 8) GABA_A_R under 1 mM GABA. (**G**) Conductance under 1 mM GABA condition. (**H**) Mean open time under 1 mM GABA condition. (**I**) Channel-opening frequency under 1 mM GABA condition. (**J**) Open probability under 1 mM GABA condition. Statistical differences were determined using one-way ANOVA (* *p* < 0.05, ** *p* < 0.01, *** *p* < 0.001, ns = not significant).

**Figure 5 ijms-23-02723-f005:**
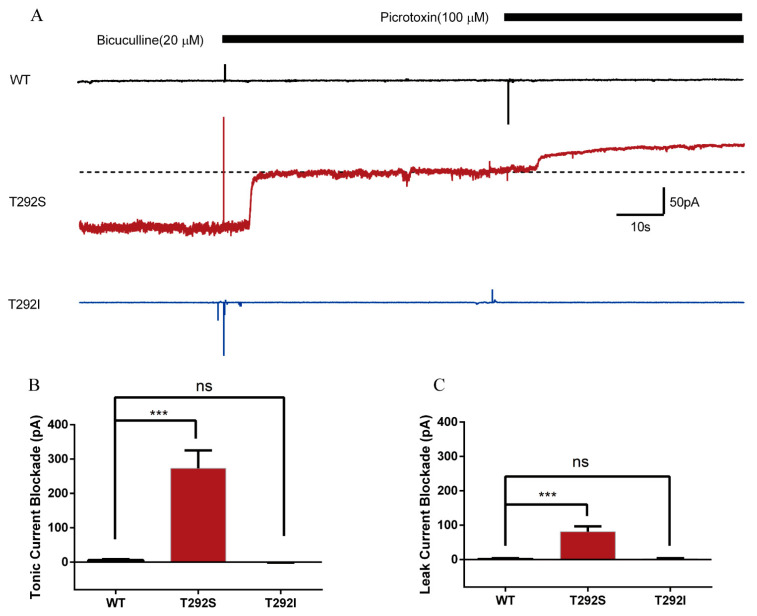
Tonic and leak current assays on WT, α1_T292S_, and α1_T292I_ of GABA_A_R show increased tonic and leak currents in the T292S mutant receptor. (**A**) Representative trace of bicuculline-blocked tonic current and picrotoxin-blocked leak current of WT, α1_T292S_, and α1_T292I_ GABA_A_R. Dotted line indicates the new baseline after tonic current fully blocked by 20 μM bicuculline for the quantification of additional 100 μM picrotoxin-blocked leak current. (**B**) Quantification of tonic current blocked by bicuculline in WT (*n* = 5), α1_T292S_ (*n* = 9), and α1_T292I_ (*n* = 8) of GABA_A_R. (**C**) Quantification of leak current blocked by picrotoxin in WT (*n* = 5), α1_T292S_ (*n* = 8), and α1_T292I_ (*n* = 7) of GABA_A_R. Statistical differences were determined using one-way ANOVA (*** *p* < 0.001, ns = not significant).

**Figure 6 ijms-23-02723-f006:**
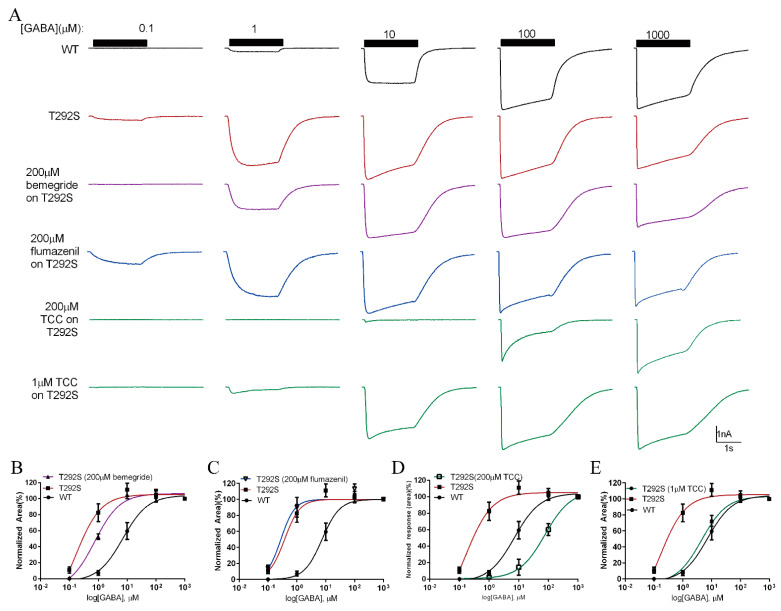
The restoring effects of the negative GABA_A_R allosteric modulators on α1_T292S_ mutant GABA_A_R. (**A**) Representative trace of GABA-evoked currents of the WT (black), α1_T292S_ (red), and α1_T292S_ GABA_A_R treated by bemegride (purple), flumazenil (blue), and thiocochicoside (TCC) (green). (**B**) Dose-response curve of α1_T292S_ GABA_A_R treated with bemegride (*n* = 5) in comparison with untreated α1T292S (*n* = 8) and WT (*n* = 10) GABA_A_Rs. (**C**) Dose-response curve of α1_T292S_ GABA_A_R treated by flumazenil (*n* = 6) in comparison with untreated α1_T292S_ (*n* = 8) and WT (*n* = 10) GABA_A_Rs. (**D**) Dose-response curve of α1_T292S_ GABA_A_R with thiocochicoside (TCC) (*n* = 8) treatment compared with untreated α1_T292S_ (*n* = 8) and WT (*n* = 10) GABA_A_Rs. (**E**) Dose-response curve of α1T292S GABA_A_R with 1 μM thiocochicoside (TCC) (*n* = 7) treatment compared with untreated α1T292S (*n* = 8) and WT (*n* = 10) GABA_A_Rs.

**Figure 7 ijms-23-02723-f007:**
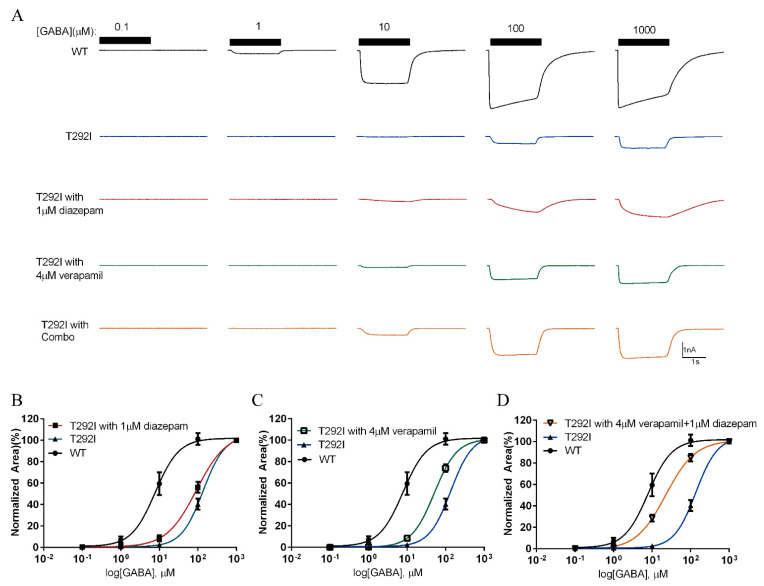
The rescuing effects of diazepam and verapamil on the α1_T292I_ mutant GABA_A_R. (**A**) Representative trace of GABA-evoked currents of WT (black), α1_T292I_ (blue) GABA_A_R, and α1_T292I_ with treatment of diazepam (red), verapamil (green), and the combination of both (orange). (**B**) Dose-response curve of α1_T292I_ GABA_A_R with diazepam (*n* = 6) in comparison with untreated α1_T292I_ (*n* = 8) and WT (*n* = 10) GABA_A_Rs. (**C**) Dose-response curve of α1_T292I_ GABA_A_R with verapamil (*n* = 7) treatment in comparison with untreated α1_T292I_ (*n* = 8) and WT (*n* = 10) GABA_A_Rs. (**D**) Dose-response curve of α1T292I GABA_A_R with combined treatments of diazepam and verapamil (*n* = 8) compared with untreated α1_T292I_ (*n* = 8) and WT (*n* = 10) GABA_A_Rs.

**Table 1 ijms-23-02723-t001:** Summary of the GABA_A_R functional alterations of the T292S and T292I variants and clinical phenotypes of the two patients carrying these two mutations. (up and down arrows indicate increase and decrease respectively).

	Patient 1 (Present Study)	Patient 2 [19,31]
Onset age	6 months (developmental delay)	3 months (epilepsy)
Detection method	Whole exome sequencing	Whole exome sequencing
Nucleotide change	NM000806.5: *c.875 C>G*	NM000806.5: *c.875 C>T*
Protein change	p.Thr292Ser	p.Thr292Ile
Inheritance	De novo	De novo
Clinical features	No diagnosed seizure eventsDevelopmental delay	Lennox–Gastaut syndrome with light-sensitive myoclonic epilepsy and generalized tonic–clonic seizures
EEG	a few abnormal discharge	N/A
MRI	Normal	N/A
Functional alterations	Agonist sensitivity 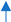 Maximum response: normal EC50 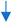 Surface/total expression: normalChannel open probability 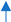 Tonic and leak current 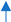	Agonist sensitivity 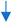 Maximum response 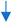 EC50 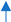 Surface/total expression: normalChannel open probability 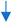 Tonic and leak current: normal

## Data Availability

Data are contained within the article or Appendix A. Constructed plasmids are available upon request from the corresponding author.

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
