# Peer review of "Distinct Functional Alterations and Therapeutic Options of Two Pathological De Novo Variants of the T292 Residue of GABRA1 Identified in Children with Epileptic Encephalopathy and Neurodevelopmental Disorders"

_ijms, 2022, doi:10.3390/ijms23052723_

Round 1

Reviewer 1 Report

My suggestions:

  1. Did the proband child have any siblings or any relatives with neurodegeneration?
  2. Was PET analysis or CT performed on the patient? If yes, did it eveal any changes?
  3. In the discussion, I would add a table, which compares the clinical phenotypes of GABRA1 T292S and T292I (age of onset, imaging data, clinical symptoms, functional studies).
  4. Were there any other similar mutations described on GABRA1 gene? It would be nice to mention briefly them in the introduction or discussion.
  5. You may mention briefly some other gentic risk factors for epilipsy in intoduction or discussion. Did you check other epilepsy risk genes? Were there mutations found in these genes?

Author Response

We would like to thank the reviewers for their constructive comments, which have directed our efforts in further strengthening the manuscript.  Please find our following responses to each comment:

Reviewer one:

  1. Did the proband child have any siblings or any relatives with neurodegeneration?

Response: We thank the reviewer for asking this very relevant question.  The child does not have any siblings or family members with neurodegeneration.  We added this information in line 103-104 of the text.

  1. Was PET analysis or CT performed on the patient? If yes, did it reveal any changes?

Response: We thank the reviewer for asking this relevant question.  Unfortunately, no PET analysis or CT scan was ever performed on the patient.

  1. In the discussion, I would add a table, which compares the clinical phenotypes of GABRA1 T292S and T292I (age of onset, imaging data, clinical symptoms, functional studies).

Response: We thank the reviewer for the great suggestion and in line with his/her suggestion, we have added a table (Table 1) to the manuscript.

  1. Were there any other similar mutations described on GABRA1 gene? It would be nice to mention briefly them in the introduction or discussion.

Response: We thank the reviewer for this important question and took the suggestion to briefly mention other GABRA1 gene mutations discovered thus far in the discussion (line 399-407) of revised manuscript. 

  1. You may mention briefly some other genetic risk factors for epilepsy in introduction or discussion. Did you check other epilepsy risk genes? Were there mutations found in these genes?

Response: We thank the reviewer for the excellent suggestion. The epilepsy risk genes were briefly introduced at the beginning of introduction (line 38-42) according to the suggestion.

A whole exome sequencing was conducted on the patient and no other mutation was found.

Reviewer 2 Report

This is an interesting paper. I just have some comments.

Major issues

Since the patient is only 2-year-old, LGS cannot be diagnosed at this timing. I guess the patient may have epileptic seizure later.

Current version is too strong. The authors may be recommended to state that the patient might have observable epileptic seizure later.

Minor issues

Early EE needs full words.

EEG needs full words.

Author Response

We would like to thank the reviewers for their constructive comments, which have directed our efforts in further strengthening the manuscript.  Please find our following responses to each comment:

Reviewer Two:

  1. Major issues

Since the patient is only 2-year-old, LGS cannot be diagnosed at this timing. I guess the patient may have epileptic seizure later.

Current version is too strong. The authors may be recommended to state that the patient might have observable epileptic seizure later.

Response: We thank the reviewer for this very good point and recommendation. We totally agree with the reviewer that we cannot rule out the possibility of the occurrence of epilepsy in the future during potential alterations of the brain circuits at late developmental stages. We have changed the statement in the result and added this possibility in the discussion (line 343-344) according to the suggestion.

  1. Minor issues

Early EE needs full words.

Response: We thank the reviewer for pointing this out and have added full words “epileptic encephalopathy” for EE accordingly (line 49).

EEG needs full words.

Response: Thanks again for the reminder; we have added full words “electroencephalography” for EEG in line 102 of the text.

Round 2

Reviewer 1 Report

Manuscript is acceptable now